# A Comparison of Different Stomatal Density Phenotypes of *Hordeum vulgare* under Varied Watering Regimes Reveals Superior Genotypes with Enhanced Drought Tolerance

**DOI:** 10.3390/plants12152840

**Published:** 2023-08-01

**Authors:** Brittany Clare Robertson, Yong Han, Chengdao Li

**Affiliations:** 1Western Crop Genetics Alliance, College of Science, Health, Engineering and Education, Murdoch University, 90 South Street, Murdoch, WA 6150, Australia; bcrobertson97@gmail.com (B.C.R.); yong.han@dpird.wa.gov.au (Y.H.); 2Western Australian State Agricultural Biotechnology Centre, Murdoch University, 90 South Street, Murdoch, WA 6150, Australia; 3Department of Primary Industries and Regional Development, 3-Baron-Hay Court, South Perth, WA 6151, Australia

**Keywords:** stomata, water-use efficiency, barley, drought tolerance, yield

## Abstract

Enhancing the water-use efficiency (WUE) of barley cultivars may safeguard yield deficits during periods of low rainfall. Reduced stomatal density is linked to enhanced WUE, leading to improved drought resistance across plant genera. In this study, 10 barley varieties exhibiting a range of stomatal density phenotypes were grown under differing soil water contents to determine whether stomatal density influences the capacity of genotypes to resist low water availability. The low-stomatal-density genotype Hindmarsh showed the least impact on biomass production during early development, with a 37.13% decrease in dry biomass during drought treatment. Low-stomatal-density genotypes additionally outcompeted high-stomatal-density genotypes under water-deprivation conditions during the reproductive phase of development, exhibiting 19.35% greater wilting resistance and generating 54.62% more heads relative to high-stomatal-density genotypes (*p* < 0.05). Finally, a correlation analysis revealed a strong negative linear relationship between stomatal density and the traits of head number (r = −0.71) and the number of days until wilting symptoms (r = −0.67) (*p* < 0.05). The combined results indicate that low-stomatal-density genotypes show promising attributes for high WUE, revealing novel barley varieties that may be useful to future breed improvement for drought tolerance.

## 1. Introduction

The maintenance and improvement of crop productivity is an essential element of global food security. However, agronomic yields are becoming increasingly impacted by climate change-induced shifts in temperature, atmospheric composition, and annual precipitation. If the climate problem is not addressed by adapting current agricultural practices, we may see global malnutrition rates rise by 20% by the year 2050 [1]. Further exacerbating the climate threat to food security is the growing demand for food supply, expected to increase by 85% by 2050 [2]. Asia, which accounts for nearly 60% of the global human population, is experiencing further population expansion, being expected to grow to 5.2 billion people by the year 2050 [3]. More alarming is the large proportion of Asian citizens with a rural lifestyle (70%) who are reliant on agricultural practices for sustainability [3]. Given that drought accounts for almost 20% of agronomic losses in such regions, this highlights the urgency to increase the adaptability of crops grown in these population dense areas—especially given that traditionally arid regions, such as Pakistan and India, are projected to experience future rises in average temperature [3,4].

In the low-rainfall country of Australia, climate change is generating increasingly erratic and unpredictable weather events defined by increased drought frequency and higher average temperatures [5]. In turn, the accompanying yield reductions have resulted in a 23% decrease in average farm profitability [6]. Overall, this effect is predicted to cost the Australian economy AUD 19 billion by the year 2030 [7]. In major cropping regions of Australia, such as the south-west of Western Australia, winter rainfall (recorded from April to October) has declined by 20% in 50 years [6]. In the south-west, annual rainfall is also predicted to reduce by 18% by the year 2090 in a high CO_2_ emissions scenario [8]. This is of particular detriment to crops growing in the wheatbelt region, which rely almost exclusively on seasonal rainfall [9]. Western Australian grain growers are additionally responsible for 40% of the nation’s barley production [10]. Given that the Western Australian climate is predicted to experience prolonged drought periods and increased temperatures, there is great emphasis on a need to enhance the drought tolerance of Western Australian barley cultivars to safeguard yield deficits during extreme weather events.

Studies have demonstrated that plants with high water-use efficiency (WUE) have greater rates of survival in low rainfall environments and produce superior yields versus lower WUE varieties [11]. WUE relates to a plant’s capacity to accumulate biomass and/or assimilate CO_2_ to its rate of water loss through transpiration [12]. The manipulation of stomatal characteristics, including stomatal density, stomatal aperture, and stomatal spacing, have been shown to have positive effects on WUE and/or photosynthetic efficacy in members of the bryophyte, eudicot, and monocot lineages [13,14,15,16,17,18]. Stomata are integral structures and are a ubiquitous feature on the epidermis of almost all land plants [19]. Stomata are the site of gas exchange, namely the diffusion of CO_2_ for photosynthetic processes. However, they also serve as the exit point for water vapour escape through transpiration. The explicit link between stomatal function and water retention has highlighted stomatal modification as a major contender for the improvement of WUE in commercial plant varieties. Reducing stomatal density vastly increases plant capacity to utilise water effectively, with limited impacts on the net photosynthetic rate or carbon assimilation. Reduced-stomatal-density rice has exhibited 27% increases in yield versus controls under water limiting conditions, with enhanced evaporative cooling at high temperatures (40 °C) [20]. Wheat varieties exhibiting lower stomatal density experience yield reductions to a lesser degree (29.28%) versus controls (33.57%) under water-limitation conditions [21]. Wheat varieties with moderate reductions to stomatal density also exhibit stable yields when exposed to water restrictions [15]. Barley exhibits a remarkable amenability to stomatal density modification, which makes it an excellent target for the improvement of cereal WUE by stomatal manipulation. In a study by Hughes et al., barley exhibiting reductions as extreme as 88% in stomatal density relative to controls experienced yield stability and improved WUE under water-limiting conditions [18]. Extending into agronomic species not included in the grass family, tomato with reduced stomatal density tolerates a range of climate change-associated environmental effects. Such benefits include increased WUE with no impact on the photosynthetic rate or dehydration avoidance, in addition to negligible yield impacts (no difference in yields), when comparing control plants to reduced-stomatal-density lines under both well-watered and water-restricted conditions [22,23].

Domestic barley (*Hordeum vulgare* L.) is a major contender in global grain markets, partly owing to its general amenability to harsh environments and wide-ranging uses as a staple in malt production and livestock supplementation [24]. Improved barley grain outputs are of particular importance to Australian grain growers, who dominate 30–40% of global malting barley exports and 20–30% of global feed grade exports [25]. Barley is widely cultivated across the world and is a primary production crop in Australia, due to its resistance to the commonly drought-prone conditions of major cropping centres, including the wheat belt [26]. As a member of the grass family, barley possesses gene suites which are specialised to resist a range of abiotic stresses, including acidic soils, drought, and heat stress [27]. Studies have shown that the expression of barley-derived genes in other crop species confers abiotic stress tolerance—an example is the transcription factor *hvWRKY38*, which confers tolerance to dehydration [28]. The high genetic diversity of barley also makes the species highly amenable to improvement by selective breeding [29]. These attributes, combined with the primary uses of barley in the beer and stock industry, stand testament to its being Australia’s second largest broadacre crop [30]. Due to its global economic relevance, barley can be considered a primary crop target for genetic improvement. Barley remains a clear target for genetic selection practices, in part due to its heavy utilisation in major industries, including beer and livestock production, in addition to its high levels of resistance to environmental constraints versus other cereal species. One example of barley’s superior environmental resistance can be observed in its capacity to withstand saline conditions at levels that exceed those of rice, corn, and wheat [31].

As highlighted, stomatal modification shows substantial promise for the improvement of the WUE of agronomically relevant crop species. A number of studies have investigated the impact of reduced stomatal density on drought resistance in cereal crops, such as barley [18,20]. However, such studies used a limited number of barley genotypes and, thus, our understanding of the responses of different genetic backgrounds with varied stomatal densities to drought conditions remains deficient. In this study, 10 barley varieties exhibiting a range of stomatal densities were grown at various levels of soil water content. Plants were subjected to two watering regimes following the attainment of the three-leaf stage and Zadok’s growth stage 49 (ZS49) to determine any potential association between stomatal density and drought resistance during early development and the reproductive phase, respectively. Superior genotypes identified in our study would serve as a potential source of low-stomatal-density germplasm for developing high WUE cultivars with excellent yield stability.

## 2. Results

### 2.1. Distribution of Average Stomatal Density across 10 Barley Varieties

Following phenotyping of the stomatal density of the fifth leaf, genotypes exhibited a range of 11 stomata/mm^2^. The varieties HB702, TR158, and Hindmarsh possessed a low average stomatal density of the fifth leaf (with an average stomatal density less than or equal to the quartile 1 (Q1) value of 16.3 stomata/mm^2^). In turn, the varieties Schooner, CDC Unity, and SVB21 were determined to possess a high average stomatal density of the fifth leaf relative to other varieties grown in the experiment, with an average stomatal density greater than or equal to the quartile 3 (Q3) value of 19.1 stomata/mm^2^. Figure 1 shows the relative average stomatal density of the fifth leaf of the 10 varieties studied, where SVB21 possessed the highest average stomatal density of 19.6 stomata/mm^2^ and TR158 possessed the lowest average stomatal density, with 14.3 stomata/mm^2^. No statistically significant difference was apparent between varieties for the trait of stomatal density on the fifth leaf.

As shown in Figure 2 below, a statistically significant difference was apparent between varieties in terms of the average flag-leaf stomatal density (*p* < 0.0001). The flag-leaf stomatal density of varieties grown to maturity in this glasshouse experiment had a large range of 63 stomata/mm^2^. Although this pattern contrasted with the more homogenous stomatal density observations taken from the fifth leaf of plants grown to the tillering stage, some similarities ensued. This is reflected in the observation that SVB21 exhibited the highest average stomatal density across both experiments, with an average fifth-leaf density of 19.6 stomata/mm^2^ and a flag-leaf density of 57.4 stomata/mm^2^ (Figure 1 and Figure 2).

As outlined previously, following the collection of fifth-leaf stomatal density measurements, Hindmarsh was considered a low-stomatal-density variety, exhibiting an average density of 16.1 stomata/mm^2^ (Figure 1). Hindmarsh similarly exhibited a low flag-leaf stomatal density, in turn producing the lowest flag-leaf stomatal density out of all 10 varieties included in the study group (33.9 stomata/mm^2^) (Figure 2). In terms of the average flag-leaf stomatal density, SVB21 differed significantly from the Hindmarsh, Schooner, Roe, CDC Meredith, and HB702 varieties (*p* < 0.05) (Figure 2). Hindmarsh exhibited a significant difference in flag-leaf stomatal density relative to the CDC Unity, SVB21, Hamelin, TR158, and VB0904 varieties (*p* < 0.05) (Figure 2).

### 2.2. Hindmarsh Least Impacted by Drought Effects during the Early Stages of Development

Figure 3 above shows the relative average fresh and dry weight produced for 10 barley varieties that were subjected to a control and drought treatment following the attainment of the three-leaf stage. All 10 varieties exhibited a significant difference between the drought and control treatments in terms of the production of both fresh and dry biomass (*p* < 0.05). Hindmarsh experienced the least impact on dry biomass during water restriction with a 37.13% decrease in dry weight, followed by SVB21 with a 45.27% reduction in dry weight versus the control (Figure 3C). Hindmarsh additionally experienced the least impact on fresh weight with an 85.22% decrease in fresh biomass during the drought treatment relative to the control (Figure 3D). Hindmarsh also demonstrated the least impact on moisture content between the drought and control treatments, with a 90.67% reduction in water content (Figure 3E). On the other hand, TR158 experienced the most substantial impact on water content under the drought conditions of all the varieties tested, with a 96.22% reduction in moisture content. TR158 also experienced the most substantial impacts on both fresh and dry biomass, with a 63.31% reduction in dry weight and a 91.32% reduction in fresh weight under the drought conditions (Figure 3C,D).

During the control treatment, CDC Meredith produced the highest average value for both the dry and fresh biomass readings, with an average value of 4.43 g and 35.4 g, respectively (Figure 3A,B). HB702 produced the lowest average dry weight (3.22 g) and CDC Unity produced the lowest average fresh weight (26.5 g) under the control conditions (Figure 3A,B). No statistically significant differences between varieties were determined for both the fresh and dry biomass collected for varieties grown under the control treatment conditions (Figure 3A,B).

Statistically significant differences between the varieties were evident for both fresh and dry weight under the drought treatment conditions (*p* < 0.0001). Hindmarsh possessed the highest average fresh weight (4.95 g) following the drought treatment, differing significantly from the varieties CDC Meredith, CDC Unity, Hamelin, HB702, Roe, TR158, and VB0904 (*p* < 0.05) (Figure 3B). SVB21 possessed the highest average dry weight (2.14 g) following the drought treatment, differing significantly from CDC Unity, HB702, Roe, TR158, and VB0904 (*p* < 0.05). Hindmarsh possessed the second highest dry weight value of 2.10 g under the drought conditions, with significant differences from HB702, Roe, TR158, and VB0904 (*p* < 0.05). VB0904 produced the lowest average value for dry weight (1.42 g) under the drought conditions, with TR158 producing the lowest average fresh weight following the drought treatment (2.77 g)—both varieties differed significantly from SVB21, Hindmarsh, and Schooner (*p* < 0.05) (Figure 3A,B).

### 2.3. A Comparison of Yield Traits at Varied Levels of Water Supplementation at the ZS49 Growth Stage

A number of trends were revealed when comparing all three watering regimes for the traits of height, shoot dry biomass, head weight, and head number. In terms of plant height, a statistically significant difference was apparent across all three treatments (*p* < 0.0001), with an average height of 63.7 cm for plants grown under the control watering regime, 51.6 cm for those under the intermediate watering regime, followed by an average of 48.2 cm for plants subjected to drought (Figure 4A). For shoot dry biomass measurements, plants under the control treatment exhibited an average biomass of 38.6 g, compared to an average biomass of 37.7 g and 30.9 g for the intermediate and drought treatments, respectively (Figure 4B). Although the average biomass for the control and intermediate treatments both differed significantly from the drought treatment, there was no significant difference in biomass between plants grown under the intermediate and control water regimes (Figure 4B). This trend was also evident when investigating the average head number between treatments, whereby plants undergoing the control treatment produced an average head number of 37.9, followed closely by the intermediate treatment average of 37 heads; therefore, the difference in averages was not statistically significant between these treatments (Figure 4D). However, the average head number dropped drastically to 4.64 for plants subjected to the drought treatment and this differed significantly from both the control and intermediate watering regimes (*p* < 0.0001) (Figure 4D). As with the height measurements, a clear distinction in the average head weight was apparent between treatments, with a statistically significant difference evident following pairwise comparisons between all three treatments (Figure 4C). Plants in the control treatment produced an average head weight of 30.5 g, outcompeting those grown under both the intermediate (average head weight = 18.2 g) and drought (average head weight = 0.796 g) regimes (Figure 4C).

Figure 5 shows the results of investigating the effect of the three different watering regimes on 10 barley varieties following the attainment of ZS49. In terms of shoot dry biomass, all the Canadian varieties (except HB702) exhibited no impact on biomass production across all three treatments (Figure 5B). Schooner was also noted to experience no impact on biomass production across treatments. Roe was most impacted in terms of biomass production under drought conditions, with a 39.67% reduction in biomass following the drought treatment versus the control (Figure 5F). All varieties demonstrated a statistically significant difference between the control and drought treatment for the traits of height (excluding CDC Unity), total head weight, and head number (Figure 5A,C,D). Schooner displayed the least reduction in head number, decreasing by 79.32% relative to the control, followed by Hindmarsh with an 80.53% decrease in head number following the drought versus the control treatment (Figure 5H). On the other hand, SVB21 was most impacted, with a 95.63% reduction in head number following drought treatment (compared with the control) (Figure 5H). The same pattern was followed once again when comparing the head weight for individual varieties across treatments, where Schooner exhibited the least impact with a 93.74% decrease versus the control under drought conditions, followed by Hindmarsh with a 95.54% decrease, with SVB21 experiencing the most severe reduction, with a 99% decrease in head weight under drought conditions relative to the control watering regime (Figure 5G). However, Hindmarsh also showed one of the greatest reductions in head number when comparing the intermediate and control watering regimes, with a 27.83% reduction in head number under the intermediate versus the control treatment conditions (Figure 5H). In addition, only Hindmarsh showed a statistically significant difference in head number between the intermediate and drought treatments out of all 10 varieties tested (Figure 5D). Hindmarsh exhibited the most stability in terms of plant height between the drought and control treatments, with a 14.31% reduction in height (Figure 5E). SVB21 showed the greatest severity in height reduction, with a 32.39% decrease in height under drought treatment conditions (relative to the control treatment) (Figure 5E).

Significant differences were evident between varieties across all traits during the drought treatment. Schooner demonstrated the greatest average height following the drought treatment (with an average height of 54.4 cm), differing significantly from the varieties SVB21 and TR158 (*p* < 0.05) (Figure 5A). Hamelin, the second tallest variety (average height = 53.3 cm), showed a significant difference in height relative to the varieties SVB21 and TR158 (*p* < 0.05) (Figure 5A). Hindmarsh also significantly differed from both TR158 and SVB21 (*p* < 0.05)*,* which possessed the lowest average height values following the drought treatment, with average heights of 38.2 cm and 43.1 cm, respectively (Figure 5A). As the shortest variety during the drought treatment, SVB21 exhibited additional significant differences in height relative to the varieties VB0904, CDC Unity, and CDC Meredith (*p* < 0.05). In terms of shoot dry biomass, CDC Meredith and TR158 produced the highest biomass measurements, with an average of 43.6 and 48.9 g, respectively (Figure 5B). CDC Meredith significantly differed in terms of average dry biomass from Hamelin, HB702, Hindmarsh, Roe, SVB21, and VB0904. TR158 differed significantly in terms of shoot dry biomass from the other varieties undergoing the drought treatment—these being HB702, Roe, VB0904, and Hamelin—which displayed an average shoot biomass of 23.6 g, the lowest of all varieties undergoing the drought treatment (*p* < 0.05) (Figure 5B).

Schooner exhibited the greatest average head weight following the drought treatment, at 1.96 g, in contrast to SVB21, which demonstrated the lowest head weight at 0.315 g (Figure 5C). Significant differences were apparent in the average head weight between varieties (*p* < 0.05), with statistically significant pairwise differences in the average head weight evident between Schooner and Roe, and Schooner and SVB21 following a post–hoc analysis. Hindmarsh by far outcompeted all the other varieties during the drought treatment in terms of the total number of heads produced, with an average of 11.2 heads per individual. This greatly contrasted with SVB21, which exhibited the lowest average head number of 1.75 heads per plant (Figure 5D). Hindmarsh was the only variety boasting a significant difference in its distribution of head count measurements and this was apparent relative to varieties CDC Unity, Hamelin, Roe, SVB21, TR158, and VB0904 (*p* < 0.01).

Schooner, with an average height of 55.8 cm, and SVB21, with an average height of 44.9 cm, were each the tallest and shortest varieties, respectively, following the intermediate watering treatment (Figure 5A). A post-hoc analysis revealed a statistically significant difference in height between SVB21 and the varieties possessing the top three average height values following the intermediate treatment—these being Hamelin (average height = 54.6 cm), CDC Meredith (average height = 55.3 cm), and Schooner (*p* < 0.05). TR158 displayed the highest average shoot dry biomass under the intermediate watering regime, of 50.4 g (Figure 5B). Only TR158 was significantly different from all the other distributions of biomass between the varieties. TR158 differed significantly from two varieties, HB702 and Hamelin (*p* < 0.05), with the two lowest average biomass readings of 33.2 and 31.9, respectively (Figure 5B). Figure 5C shows the distribution of the average head weights following the intermediate treatment. Only one significant difference in head weight was found following the intermediate treatment, that being between CDC Meredith, with an average head weight of 23.8 g, and HB702 with an average head weight of 12.4 g (*p* < 0.05). CDC Meredith and HB702 exhibited the highest and lowest head weights, respectively, out of all the varieties during the intermediate treatment (Figure 5C). There were no statistically significant differences in head number for varieties grown under intermediate water supplementation conditions.

The minimum and maximum values of average plant height under the control watering regime mirrored those of both the intermediate and drought treatments, with SVB21 exhibiting the lowest height at 56.5 cm, and Schooner producing the highest average height at 72.8 cm (Figure 5A). Schooner was found to differ significantly in height from the varieties SVB21, TR158, VB0904, and Hindmarsh during the control treatment (*p* < 0.05). One additional variety pair demonstrated a significant difference in average height, that being SVB21 and Hamelin (*p* < 0.05). During the control treatment, no statistically significant differences were found between varieties for both the traits of head weight and shoot dry biomass. Reflecting the results of the drought treatment, Hindmarsh exhibited the highest average head number of 57.5 heads per individual during the control watering treatment (Figure 5D). Hindmarsh was the only variety to exhibit a statistically significant difference in head number relative to other varieties undergoing the control treatment. The number of heads of Hindmarsh differed significantly from the varieties CDC Meredith, CDC Unity, HB702, Schooner, TR158, and Hamelin (*p* < 0.05).

To better understand the mechanisms of drought tolerance for varieties undergoing the intermediate and drought treatments, the number of days until the first signs of wilting (following treatment initiation) were recorded for all individuals. A statistically significant difference between varieties in terms of time until wilting symptoms was apparent at *p* < 0.0001. Figure 6 shows all the pairs that demonstrated a statistically significant difference in the average time until wilting. Hindmarsh lasted the greatest number of days without water supplementation, with an average of 7.25 days until wilting signs were exhibited (Figure 6). The average number of days until wilting for Hindmarsh differed significantly relative to the varieties CDC Unity, Hamelin, Roe, SVB21, HB702, TR158, and CDC Meredith (*p* < 0.05) (Figure 6). VB0904, with the second highest average number of days until signs of wilting (6.25 days), also showed significant differences relative to the varieties SVB21, TR158, Roe, Hamelin, CDC Meredith, and CDC Unity (*p* < 0.05). SVB21 demonstrated the least amount of time taken until signs of wilting, averaging only 3 days until wilting was evident (Figure 6). The average time until wilting for SVB21 differed significantly from the varieties CDC Meredith, Hamelin, HB702, Hindmarsh, Roe, Schooner, and VB094 (*p* < 0.05).

### 2.4. Association between Stomatal Density and the Traits of Head Number and Wilting Time across Barley Genotypes under Drought Conditions

To investigate whether any potential associations existed between the yield-related traits studied and the stomatal density of barley subjected to drought treatment at ZS49, barley genotypes were placed into two groups based on their average stomatal density based on the quartile values of the study population—high stomatal density (*n* = 31) and low stomatal density (*n* = 31). High-density genotypes possessed a stomatal density less than or equal to the Q1 value of 42 stomata/mm^2^, whereas those possessing a high average density were greater than or equal to the Q3 value of 51 stomata/mm^2^. As shown in Figure 7 above, following comparisons between the low- and high-stomatal-density groupings the traits of head number and height exhibited statistically significant differences. Low-stomatal-density varieties exhibited the greatest average head number (6.39 heads), on average producing 54.62% more heads than the high-stomatal-density grouping (average head number = 2.9) (*p* < 0.01). Low-stomatal-density varieties additionally exhibited 19.35% greater wilting resistance when compared to their high-stomatal-density counterpart. The average wilting time for low-stomatal-density genotypes was 5.48 days, and 4.42 days for high-density genotypes (*p* < 0.05). Low-stomatal-density varieties were taller on average, with an average height of 48.7 cm, versus genotypes with a high stomatal density, with an average height of 45.3 cm. The same trend was followed for the trait of head weight, with low-density genotypes producing an average head weight of 0.792 g, versus the 0.515 g average of high-stomatal-density genotypes. It should also be noted that the differences in the distributions of head weight and height between the low- and high-stomatal-density genotype groupings were near statistical significance, with *p*-values of 0.0544 and 0.0543 for each trait, respectively. No statistically significant difference was evident between low- and high-stomatal-density genotypes under drought conditions for the trait of shoot dry biomass.

The relative change between the drought and control treatment for the traits of height, head weight, head number, and biomass, in addition to the average number of days until wilting symptoms, were used in combination with the average flag-leaf stomatal density values collected for 10 individual varieties to perform a linear correlation analysis. As shown in Figure 8A,C, the traits of height and head weight demonstrated a moderate negative linear correlation with stomatal density, yet the relationship was not statistically significant. Head number showed a strong negative linear correlation with stomatal density under drought conditions, with an r value of −0.71 and the relationship was statistically significant (Figure 8D). Wilting time also exhibited a moderate-to-strong negative linear correlation with stomatal density (r = −0.67; *p* = 0.036) (Figure 8E). Finally, biomass showed no association with stomatal density under drought conditions (Figure 8B).

## 3. Discussion

### 3.1. Impacts on Water-Use Efficiency May Reduce with Plant Size for Barley

SVB21 showed one of the lowest impacts on dry biomass under water limiting conditions, which was only outcompeted by Hindmarsh. These observations were noted to conflict with the fact that SVB21 produced the highest average stomatal density following sampling of the fifth leaf and flag leaf. Given this result, we must first consider the influence of high-stomatal-density phenotypes on enhanced photosynthetic capacity. Sakoda et al. showed that increased stomatal density in *Arabidopsis* improved photosynthetic efficiency leading to improved biomass production, yet it should also be noted that such individuals were not exposed to drought conditions, which likely would have impacted the capacity of high-stomatal-density phenotypes to thrive as effectively [32]. Alternatively, the fact that SVB21 managed to produce a high dry biomass result under conditions of minimal water supervision despite possessing the highest average stomatal density, potentially suggests that the enhanced capacity of SVB21 to generate biomass under drought conditions was instead a result of alternative genetic mechanisms. SVB21 showed a prostrate growth habit, exhibiting the lowest height across all three watering treatments when grown to maturity in this experiment. Multiple studies have shown an association between prostrate growth habits and polymorphisms in semi-dwarf genes, including *sdw-1* [33]. The prostrate growth habit of SVB21 could, thus, potentially suggest an association with semi-dwarf genes for this variety. A recent study investigated the impact of a lack of function mutants for the gene *chiquita1*, which produces dwarf phenotypes in *Arabidopsis thaliana*, and found that semi-dwarf phenotypes, despite showing no improvement in drought tolerance, did exhibit a higher water-use efficiency versus controls [34]. Hence, a potential association between a prostrate growth habit in SVB21 and the capacity for biomass production during limited water availability is possible, yet it does not account for the reduced viability of SVB21 during the reproductive phase under drought treatment conditions in our study. Given that *Arabidopsis* is comparatively small upon reaching maturity and immature barley grown to the tillering stage is relative in proportion, it may be postulated that impacts on water-use efficiency may only become substantial in terms of yield attributes once certain size limits are reached during development [35]. This may be a result of increases in energy demand as the plant enters the reproductive phase [36]. Somewhat defending this assertion is a study performed by Blum et al., where the authors showed that wheat mutants with a smaller overall size were more stress resistant, due to their reduced energy requirements as a result of their slow growth rate [37]. This could in turn explain the poor performance of the yield-related traits exhibited by SVB21 under drought conditions following the reproductive phase, versus the lack of impact on biomass production for SVB21 subjected to drought conditions in the early phases of development in this study.

### 3.2. Hindmarsh Exhibits Superior Biomass-Production and Water-Retention Traits under Drought Conditions during Early Development

Under drought conditions, Hindmarsh produced the highest fresh biomass of all varieties grown to the tillering stage. Considering that fresh biomass incorporates water content, this was suggestive that Hindmarsh possessed a high capacity for WUE—this assertion is further bolstered by its relatively low stomatal density in comparison to other varieties in our experiment. In addition, despite SVB21 producing the highest dry biomass reading following the drought treatment at the three-leaf stage, Hindmarsh showed the lowest decrease in dry biomass out of all the varieties, additionally exhibiting the least impact on fresh biomass under drought conditions and the lowest reduction in water content between the drought and control treatments. The combined high performance of Hindmarsh when comparing both fresh and dry biomass measurements may potentially stand testament to its consistently low values of stomatal density and, therefore, reiterate the suitability of low-stomatal-density varieties for future breeding programs. Low-stomatal-density varieties, such as Hindmarsh, may be particularly suitable for regions where there is a high risk of reduced rainfall during the vegetative growth phase, considering that Hindmarsh was most amenable to drought impacts in the early phases of development in this study. Utilising such adaptable genotypes when developing new varieties is especially important considering that the risk of extreme drought in important cropping regions (such as the south-west of Western Australia) is projected to be significantly more extreme than previously estimated [38].

### 3.3. Barley Height and Head Weight Show Increased Sensitivity to Changes in Water Supplementation during the Reproductive Phase versus Head Number and Shoot Biomass in This Study

There was a clear separation in the distribution of measurements between all three watering treatments for all traits studied, with the average value increasing for the given trait with increased water supplementation. Only height and head weight showed statistically significant differences between all three treatments, thus indicating that these two traits showed greater sensitivity in response to changes in water availability during the reproductive phase of barley development. This suggested that even moderate reductions in water content affect grain-filling capacity. This observation correlates with the general understanding that grain filling is a highly energy-intensive process, with grain-filling success being highly sensitive to fluctuations in environmental circumstances, including soil-moisture content [39].

Plant height has been previously shown to vary gradually with fluctuations in water content [40]. However, only TR158 and SVB21 showed a statistically significant difference in plant height between the drought and intermediate treatments, indicating that for some barley genotypes, the effect of water availability on plant-height variability is reduced at lower soil-water concentrations. This assertion is reflected in the findings of a recent study, showing that shoot competition between plant genotypes under low water-availability conditions was weaker than under high water-availability conditions, owing to the enhanced root competition under low water-availability conditions [41]. Thus, a change in the plant’s allocation of resources towards root formation at lower water levels may explain the lack of a statistically significant difference in height between the intermediate and drought treatments for the majority of varieties in this study [41].

### 3.4. Low-Stomatal-Density Variety Hindmarsh Outcompetes High-Stomatal-Density Variety SVB21 across Multiple Yield-Associated Traits under Both Low and High Water Supplementation

Schooner’s superior capacity for grain filling under conditions of extreme water deficit is notable in our experiment, showing both the highest grain weight under drought conditions and the lowest decrease in grain weight between the drought and control conditions. This may in part be attributed to the intermediate flag-leaf stomatal density observed for Schooner, and/or genetic mechanisms pertaining to height, considering that Schooner was the tallest variety across all three treatments (*p* < 0.001). Plant height has been shown to be an excellent indicator of growth capacity and, thus, yield outcomes [42]. Studies of reduced height (*Rht*) alleles in wheat have shown a negative association between height and grain quality, with dwarf and semi-dwarf mutants exhibiting a reduced grain weight versus wildtype controls [43]. Another study, involving the investigation of rice phenotypes overexpressing *OsBRI1*, resulted in an increase in plant height and a corresponding increase in grain length, leading to increases in yield [44]. Since BRI1 is a known regulator of stomatal density, there is potential that height is linked to stomatal formation in barley and, thus, associated with improved grain quality and yield under drought stress—at least this may be the case for Australian genotypes like Schooner and its relatives [45].

Interestingly, Canadian varieties were superior in terms of biomass production, with TR158 and CDC Meredith exhibiting the highest measurements for total shoot dry biomass following the drought treatment and CDC Meredith also producing the highest head weight following the intermediate and control treatments. In addition, TR158 consistently produced the greatest average biomass at all levels of water supplementation. Three out of four of the Canadian genotypes (TR158, CDC Meredith, and CDC Unity) were also robust against biomass impacts under water-limiting conditions following the ZS49 growth stage, with no significant differences in biomass evident between the control, intermediate, and drought watering treatments. Our observations were somewhat echoed in a study of drought and heat resistance in Canadian barley varieties (including CDC Meredith). Mahalingam showed that although major impacts on grain yields were apparent following combined heat and drought treatments, such varieties did not exhibit major impacts on biomass production, with CDC Meredith showing no statistically significant reduction in biomass under stress [46]. The results of this study by Mahalingam also support our observations of TR158 being capable of consistently outcompeting all other varieties in our trial regarding biomass production, including under water-restriction conditions. The study may additionally explain why CDC Meredith outcompeted all the other varieties in our trial for total head weight under intermediate and control water-supplementation conditions, yet not under drought conditions, since the authors identified impacts on seed development in Canadian varieties under water-restriction conditions [46].

As covered previously, it has been widely established that low stomatal density has been shown to influence drought tolerance across members of the grass family, with low-stomatal-density phenotypes associated with high water-use efficiency [18,20,47]. This association was especially prevalent for varieties exhibiting stomatal densities at the extreme ends of the spectrum in this study—these being Hindmarsh and SVB21. As highlighted throughout this study, Hindmarsh produced the lowest flag-leaf stomatal density, with an average stomatal density of 33.9 stomata/mm^2^. Hindmarsh exhibited a superior capacity for water retention under drought conditions when grown to the tillering stage. In addition, Hindmarsh was the most wilting-resistant variety in this study, lasting 58.63% longer until wilting symptoms were prevalent versus SVB21 (*p* < 0.0001), and producing head numbers that were 84.38% higher than SVB21 following the drought treatment (*p* < 0.0001), and 51.66% higher than Schooner under the control watering treatment conditions (*p* < 0.0001). Despite Schooner being the tallest variety across all three treatments, the height of Hindmarsh was most robust to low water availability, experiencing the lowest reduction in height under drought conditions out of all 10 varieties investigated. Hindmarsh was also robust against water deprivation for the traits of head number and head weight, being outcompeted only by Schooner under drought conditions. With these features in hand, Hindmarsh may hold the potential to be a genetic source for the improvement of water-use efficiency.

In contrast, SVB21 produced the highest average flag-leaf stomatal density across both experiments (when sampled at both the fifth leaf and the flag leaf). In accordance with what has typically been demonstrated for plants possessing high stomatal densities, SVB21 exhibited the shortest time until wilting symptoms were exhibited and consistently showed the lowest values for head weight and head number under drought conditions. Finally, SVB21 by far experienced the most impact on yield-related traits under water restriction conditions out of all 10 varieties investigated, with reductions of 32.39% in height, 99% in head weight, and 96.63% in head number under drought conditions relative to the control watering regime. Despite SVB21 showing the lowest height across all three treatments, this was determined to be a likely result of genes responsible for a prostrate growth habit typical of the SVB21 variety rather than its high stomatal density in this experiment.

### 3.5. Low Stomatal Density May Have a Positive Impact on Head Development and Wilting Resistance in Barley under Water-Limiting Conditions

An association between stomatal density with the traits of head number and number of days until wilting was revealed following the combined statistical analysis of barley individuals grouped based on their stomatal density classification, and a correlation analysis of genotypes based on their stomatal density in relation to the relative change in yield-associated traits under drought conditions. This, in turn, suggested that low rates of stomatal formation positively contribute to head development under drought conditions, in addition to contributing to the delayed onset of drought-associated symptoms. No correlation was indicated between stomatal density and biomass, and only a moderate negative correlation was apparent for the traits of height and head weight under drought conditions. The absence of a correlation between biomass and stomatal density under drought conditions during the reproductive phase was not unordinary and was likely due to the fact that the majority of biomass production in barley and other plants is generally completed prior to flower development [48].

For the traits of height and head weight, it may be postulated that alternate genes for drought tolerance that do not impact stomatal formation may be a dominant factor in determining improved yields under conditions of water restriction. It is known that a suite of genes contributes to a plant’s capacity for drought tolerance, including genes affecting flowering time, membrane and enzyme stabilisation (e.g., dehydrins), and hormonal signalling (e.g., abscisic acid receptors) [49,50,51]. Indeed, it has been shown that particular genotypes have drought-related genes that are more sensitive to fluctuations in abiotic stress factors. For instance, pyrabactin resistance-like (PYL) proteins are specialised receptors known to be one of the most substantial factors in abscisic-acid-pathway initiation in response to abiotic stress. In some genotypes of drought-tolerant cotton (*Gossypium hirsutum* L.), a hallmark feature is enhanced GhPYL9-11A-receptor expression [52]. On the other hand, the drought tolerance of the Okra leaf cotton (*Gossypium barbadense*) is instead attributed to its specific leaf architecture that improves its photosynthetic efficiency under drought conditions via a reduced leaf area [53]. Additional drought tolerance mechanisms linked to trichrome formation genes have since been identified following the study of a hybrid population derived from *G. barbadense* and *G. hirsutum* [54]. Among the cereal crops, Tibetan barley possesses an interesting genetic background, one which has adapted to the extreme cold and combined aridity of the Tibetan landscape [55]. In turn, a suite of genetic mutations has ensued in wild Tibetan genotypes to enhance their survivability. One such study investigating candidate genes for drought tolerance revealed a high level of genetic variability across wild Tibetan genotypes, and identified markers associated with genes involved in the synthesis of ABA precursors, the regulation of chlorophyll content, and stomatal aperture regulation [56]. 

Although domestic barley is known to have significantly reduced genotype variation versus wild barley progenitors, future studies investigating the variation of drought-tolerance mechanisms would be beneficial to determine potential sources of genotypic variation between agronomically relevant domestic barley varieties that may be contributing to observed drought tolerance [56]. Schooner, a variety that produced an intermediate average flag-leaf stomatal density in this experiment, exhibited a robustness to water deprivation, showing the lowest reduction in head number and head weight, following exposure to drought conditions at the ZS49 stage in this study. Hindmarsh, possessing the lowest value of stomatal density, additionally exhibited drought-resistant attributes in this study, producing the highest number of heads under both control and drought conditions and the most resistance to wilting symptoms. Given these observations, performing future investigations of the genetic mechanisms influencing the drought tolerance of Australian genotypes, perhaps through the analysis of hybrid populations of Schooner and Hindmarsh, may be beneficial to better understand the variability in genetic mechanisms influencing the drought tolerance of varieties adapted to the low-rainfall environments of Australian cropping systems.

## 4. Materials and Methods

### 4.1. Glasshouse Experiment for the Investigation of Barley-Yield Response under Various Watering Regimes

Ten varieties were selected for the investigation of stomatal density and drought tolerance, which included five Australian accessions (VB0904, Hamelin, Hindmarsh, Schooner, and Roe), four Canadian accessions (CDC Meredith, TR158, HB702, and CDC Unity) and one African accession (SVB21). The varieties were selected to represent a range of geographic origins, in order to maximise the potential to observe variation in stomatal density as a result of prospective adaptations to the climate of the respective regions [57]. In order to establish a consistent association between low stomatal density and increased WUE in barley, two glasshouse experiments were performed in tandem at the Grains Precinct, Murdoch University, Western Australia. The barley accessions were planted on the 11 May 2022 for both experiments and arranged in a random block design for the reduced likelihood of systemic error. The plants were grown in a controlled glasshouse environment and maintained within a temperature range from 19 °C to 24 °C by an EnviroSTEP integrated-control climate management system. Pots were relocated to a different space within the glasshouse chamber every week to prevent any potential microclimate effects on the phenotypes.

### 4.2. Experimental Setup for Accessions Grown to the Tillering Stage

Due to the short timeframe of this experiment, the barley lines were grown in 90 mm olive P90OPX pots, using one level 5 Ml measuring spoon of NPK (Blue) granular fast-release fertiliser and one level 10 Ml measuring spoon of Osmocote^®^ slow-release granular fertiliser manufactured by The Scotts Company LLC and sourced from Perth, Australia. Each pot comprised two individual plants. The barley plants were grown to the tillering stage, or Zadok’s stage 15 (ZS15), characterised by the emergence of 5–7 leaves on the main stem [58]. Two watering treatments were implemented. In the control treatment (*n* = 8 per variety), plants were watered every two days. For the drought treatment (*n* = 4 per variety), water was withheld completely once individuals reached the three-leaf stage. The barley lines were grown to the tillering stage for a period of 49 days (May–June 2022).

### 4.3. Experimental Setup for Accessions Grown to the ZS49 Stage

The barley was grown to Zadoks stage 49 (ZS49), characterised by a 50% awn emergence above the flag leaf [48], after which exposure to different watering regimes commenced. Larger 200 mm Slimline P200SLTL pots were used in this experimental design to accommodate for maintaining mature plants. To account for the differing phenology between varieties, individuals grown under all treatments were watered as normal (once daily) until ZS49 was reached. The individual was then tagged and monitored for soil moisture content using an ML3 ThetaKit IC ML3 moisture meter between waterings. The soil moisture content was measured using the top 6 cm of soil. Individuals were placed in three treatment groups, with eight biological replicates per variety per treatment. The soil moisture content for varieties in the control group was maintained within the potting soil field capacity (22 ± 4%) for the entirety of the experiment. The field capacity of the soil was determined by thoroughly watering the soil until saturated and then recording a soil moisture reading 48 h post-watering. Those in the intermediate treatment group were only watered once the soil moisture content fell below 11% (i.e., below half of the field capacity) and maintained a moisture level of 11 ± 5%. Water was withheld for plants subjected to the drought treatment once ZS49 was reached. The date of ZS49 was recorded for each variety, in addition to the number of days taken until the first signs of wilting and the soil moisture content at wilting for the intermediate and drought treatment groups from the day that the water treatment commenced (i.e., from the date of ZS49). Plants were determined to have entered the wilting phase once the majority of leaves exhibited flaccidity.

### 4.4. Phenotyping of Stomatal Density

For barley subjected to drought treatment at the ZS49 growth stage, stomatal density phenotyping was performed using the fully expanded flag leaf. Three flag leaves of roughly the same proportions were collected from four individuals from each variety and coated in a layer of clear nail varnish spanning approximately 1.5 cm on the adaxial surface from the base of the leaf. The nail polish layer was removed once dry with sticky tape and placed on a glass microscope slide. The stomatal density (number of stomata per mm^2^) was then recorded as an average number of stomata counted over three ‘views’ or separate portions of the leaf surface as viewed at 200× magnification, forming three technical replicates per leaf sample. The average stomatal density for each variety was finally obtained by averaging the measurements obtained for each of the three leaf samples, for each of the four replicates per variety.

For barley subjected to drought treatment at the three-leaf stage, the fifth leaf (fully expanded) was sampled from each variety in order to determine the average stomatal density present on the leaves of immature plants for each variety. The fifth leaf was sampled from four different individual plants per variety, and stomatal phenotyping was performed as described previously (i.e., slides were created using a clear nail varnish layer spanning 1.5 cm from the leaf base and stomata were counted at 200× magnification). The average stomatal density per individual plant was determined by averaging the number of stomata counted over four views of the leaf, and the average stomatal density for each of the four leaves was calculated to determine the average stomatal density per variety. Phenotyping was performed using a Saxon Researcher NM11-4100 Biological Microscope, manufactured by Saxon Optics Australia and sourced from Perth, Australia. Iand imaging of the leaf surface was completed with a Dino-Eye Edge AM7025X camera manufactured by Dino-Lite, Taiwan and sourced from Perth, Australia.

### 4.5. Phenotyping of Yield-Associated Traits

For both drought tolerance experiments, biomass was collected by cutting the stems at the very base of the plant at the level of the soil. Weight measurements were collected using both fresh and dry biomass for individuals grown to the tillering stage. Fresh biomass was recorded immediately following biomass collection. Dry samples were created by placing the fresh plant material into an oven at 45 °C for a period of 7 days. Only dry biomass was recorded for barley grown to the ZS49 stage—in this instance, samples were collected following the end of the plant life cycle wherein the drying process had occurred naturally. Measurements of all biomass readings were recorded using an AND HF-2000G scale.

Additional measurements for head number, plant height, and head weight were recorded for barley grown to the ZS49 stage. Head number was determined by counting the number of fully emerged/mature heads on each individual. Plant height (cm) was determined by measuring from the surface soil to the point of the plant where the majority of heads had developed. Finally, head weight was measured by the same means as the recording of the biomass measurements, using the same scale. Heads were collected from individual plants by cutting at the base of the mature head, at the point of attachment to the stem.

### 4.6. Statistical Analysis of Trait Data Collected from Both Glasshouse Experiments

To assess the relative differences in the distribution of measurements for the traits of fifth-leaf stomatal density, fresh biomass and dry biomass for varieties subjected to drought treatment at the three-leaf stage, and the traits of flag-leaf stomatal density, plant height, shoot dry biomass, head number, head weight, and the average number of days until wilting for varieties subjected to three watering regimes at ZS49, a one-way ANOVA was performed for individuals subjected to the different treatments accordingly. Post-hoc tests were completed for any instances where a statistically significant difference was determined in order to identify variety pairs demonstrating a significant difference.

Individuals grown in three treatment groups based on water supplementation following ZS49—the control (*n* = 80), intermediate (*n* = 76), and drought (*n* = 67) groups—were interrogated using a one-way ANOVA analysis to determine any significant pairwise differences between each of the three watering regimes for the traits of plant height, shoot dry biomass, head number, and head weight. Individual varieties were also investigated to determine if a statistically significant difference in the traits studied was apparent between watering treatments for the given variety—this was determined with a student’s *t*-test for plants subjected to drought treatment at the three-leaf stage, and one-way ANOVA for varieties subjected to three watering regimes at growth stage ZS49.

The relative change in fresh and dry biomass following the drought treatment (versus the control treatment) was calculated using the average values obtained for varieties subjected to drought treatment at the three-leaf stage using the following formula: relative change = value following drought treatment − value following control treatment/value following control treatment. The relative change in moisture content between the control and drought treatments was additionally calculated for barley subjected to drought treatment at the three-leaf stage. The relative change in moisture content was calculated by subtracting the water content (fresh weight − dry weight) under the drought conditions from the water content under the control conditions, and then dividing this value by the water content under the control conditions. For varieties subjected to three watering regimes at the ZS49 stage, the relative change between the control and drought treatment and the control and intermediate treatment was calculated for the traits of height, shoot dry biomass, total head weight, and total head number, respectively.

Varieties were classified as possessing a high stomatal density if the average stomatal density was greater than or equal to the quartile 3 (Q3) value (in the highest 75% of stomatal density measurements) and a low stomatal density when the average density was less than or equal to Q1 (in the lowest 25% of stomatal density measurements), when taking the distribution of average stomatal density across all 10 varieties into account for those grown to the tillering stage and those grown to maturity, respectively. To determine whether a statistically significant difference was apparent between high- and low-stomatal-density genotypes for the traits of height, shoot dry biomass, total head weight, total head number, and average number of days until wilting, the barley genotypes were grouped into a high-stomatal-density or low-stomatal-density category based on the average stomatal density of the flag leaf, and the distribution of measurements collected for such genotypes for the aforementioned traits were compared using a student’s *t*-test.

A correlation analysis was performed using the average stomatal density measurements collected for 10 varieties subjected to the drought treatment at the ZS49 growth stage. The Pearson correlation co-efficient (r) and its associated *p*-value was determined for the average stomatal density values (number of stomata/mm^2^) for each variety in relation to the relative change (between the control and drought treatment) for the traits of height, head weight, shoot dry biomass, and head number. A correlation analysis was also performed to determine the relationship between stomatal density and the average number of days until wilting for the 10 varieties. Scatterplots were additionally generated and fitted with a linear regression line for each trait in relation to the average stomatal density values recorded for each genotype.

All the statistical analyses described above were performed using RStudio version v1.4.1103-4 Wax Begonia. The rstatix package (v0.7.2; Kassambara, 2023) was used for the preliminary statistical assessments (e.g., confirmation of assumptions) and the statistical tests performed for the above phenotype data collected for the 10 varieties across both experiments. ggplot2 (v3.4.2; Wickam, 2016) was used to create the bar plots, violin plots, and scatterplots generated from the phenotype data collected for each experiment during the respective treatments [59].

## 5. Conclusions

The combined associations between the varieties exhibiting a range of stomatal density phenotypes in this study correlates with what is generally understood with regards to the influence of low stomatal density on drought resistance. This was reflected by our observations of the high-stomatal-density variety SVB21 demonstrating a drought-sensitive phenotype following drought treatment at ZS49, and the low-stomatal-density variety Hindmarsh exhibiting a superior capacity for drought tolerance both during the early and late stages of development. Further bolstering our confidence in low-stomatal-density genotypes was the finding that stomatal density showed a strong negative correlation with the traits of head number and number of days until wilting under drought conditions, and that low-stomatal-density genotypes outcompeted high-stomatal-density phenotypes for both of these traits under drought conditions. Our study also revealed Schooner as a promising genotype for enhanced drought tolerance, given its superior head weight production under drought conditions and its robustness to head weight reduction under drought conditions. Finally, we observed that Canadian barley genotypes generally experienced virtually no impact on biomass production under low water supplementation conditions at the ZS49 growth stage, and this was consistent with the findings of a previous study. The overall results of our study have revealed the low-stomatal-density genotype Hindmarsh as a strong candidate for potential breed improvement for drought tolerance due to the observed resistances of this variety in low-water environments.

## Figures and Tables

**Figure 1 plants-12-02840-f001:**
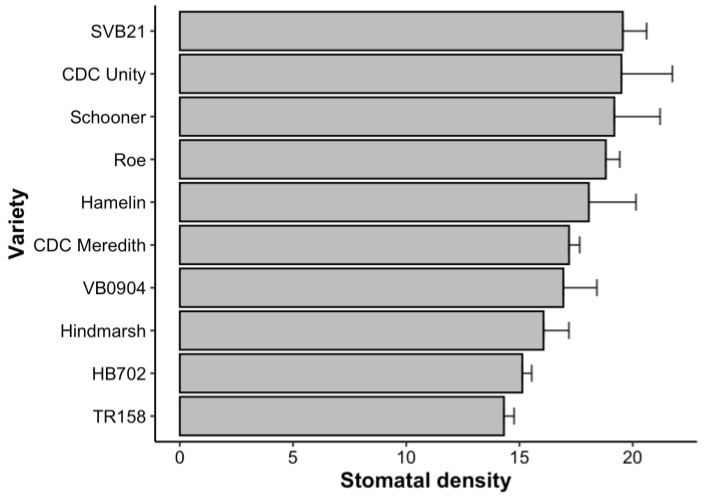
The average stomatal density of 10 barley varieties. The fifth leaf at the tillering stage was recorded. Stomatal density was the number of stomata/mm^2^ of leaf area. Error bars indicate the standard error of the mean (*n* = 4 per variety).

**Figure 2 plants-12-02840-f002:**
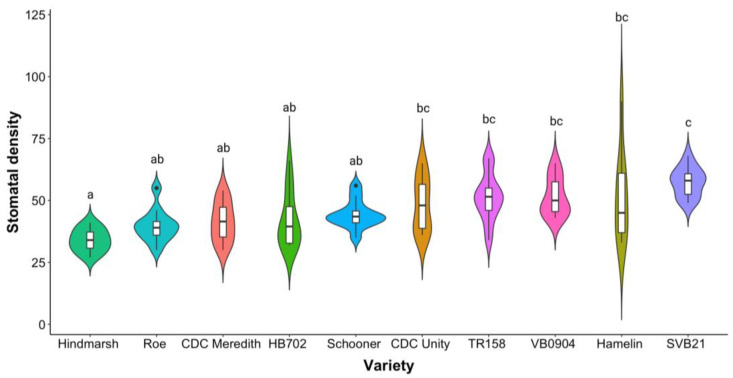
Violin plot showing the distribution of flag-leaf stomatal density observations for varieties subjected to various levels of water supplementation following the ZS49 growth stage. The average stomatal density was defined as the average number of stomata/mm^2^ on the leaf surface. Varieties that significantly differed from one another are denoted by different letters, whereas those that did not significantly differ are denoted by the same letters. Significance was attributed at *p* < 0.05.

**Figure 3 plants-12-02840-f003:**
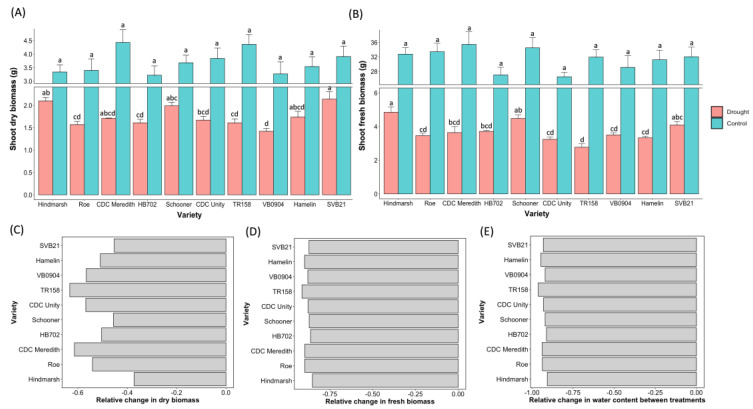
Panel figure depicting the average values for dry weight (**A**) and fresh weight (**B**) for 10 barley varieties subjected to drought and control watering treatments. Included in plots A and B are standard error bars for each treatment for the respective varieties. Varieties that significantly differ from one another under the respective treatments are denoted by different letters, whereas those that do not significantly differ are denoted by the same letter. Statistical significance was attributed at *p* < 0.05. Axis breaks have been implemented for better clarity of the phenotypic differences between varieties in the same treatment. The relative change in dry biomass (**C**), fresh biomass (**D**), and water content (**E**) between the control and drought treatments is plotted above for 10 varieties subjected to drought treatment at the three-leaf stage.

**Figure 4 plants-12-02840-f004:**
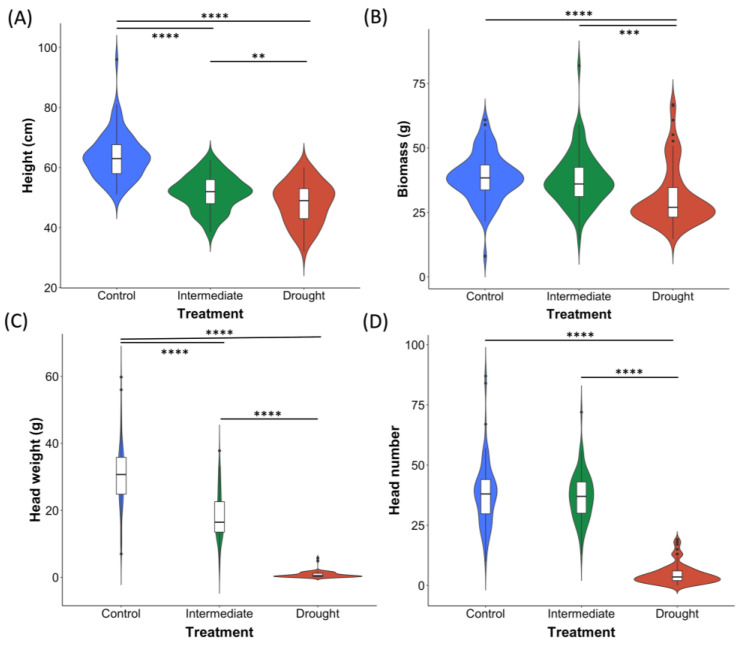
Panel figure showing the relative distributions of measurements for the traits height (**A**), biomass (**B**), head weight (**C**), and head number (**D**) for barley grown at three levels of water supplementation—drought, intermediate, and control. Plants were subjected to the respective watering regimes upon reaching the ZS49 growth stage. Also provided are notations for any treatment groups for which there was a significant difference in the distribution of measurements. Significance levels are denoted by the following: ** = *p* < 0.01; *** = *p* < 0.001; **** = *p* < 0.0001.

**Figure 5 plants-12-02840-f005:**
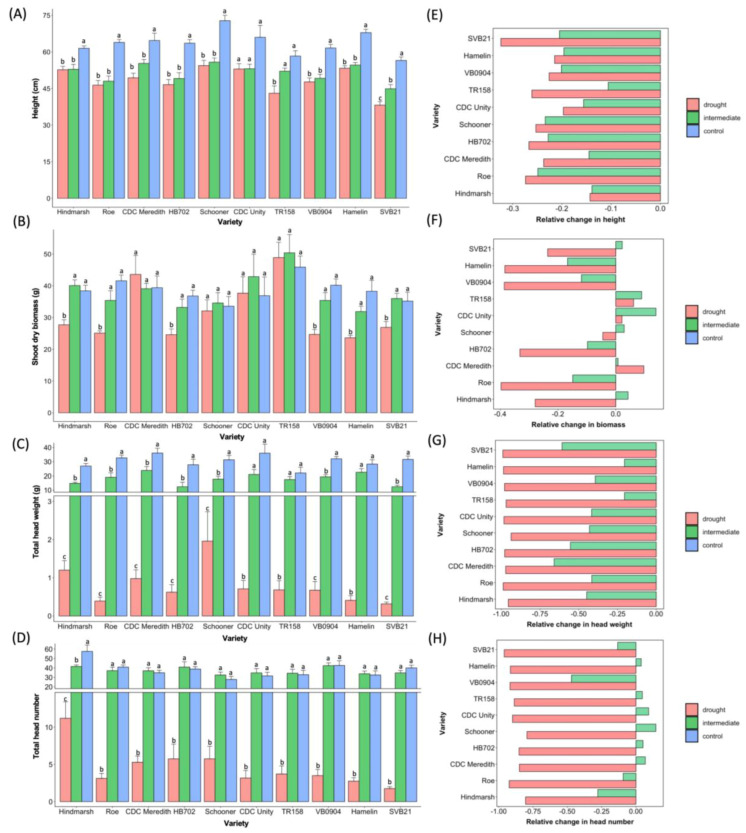
Panel figure depicting the average values for height (**A**) shoot dry biomass (**B**), total head weight (**C**), and total head number (**D**) for 10 barley varieties subjected to the drought, intermediate, and control treatments depicted in the legend above. Included in plots (**A***–***D**) are standard error bars for each treatment for the respective varieties. Axis breaks of values have been implemented for better clarity of the phenotype differences between varieties subjected to the same treatment for plots (**C**,**D**). Different letters are presented where the trait differed between treatments for a given variety, whereas those varieties with traits that did not significantly differ between treatments are denoted by the same letter. Statistical significance was attributed at *p* < 0.05. The relative change in height (**E**), shoot dry biomass (**F**), total head weight (**G**), and total head number (**H**) between the control and drought treatments for 10 varieties subjected to drought or intermediate treatment at the ZS49 growth stage are also plotted above.

**Figure 6 plants-12-02840-f006:**
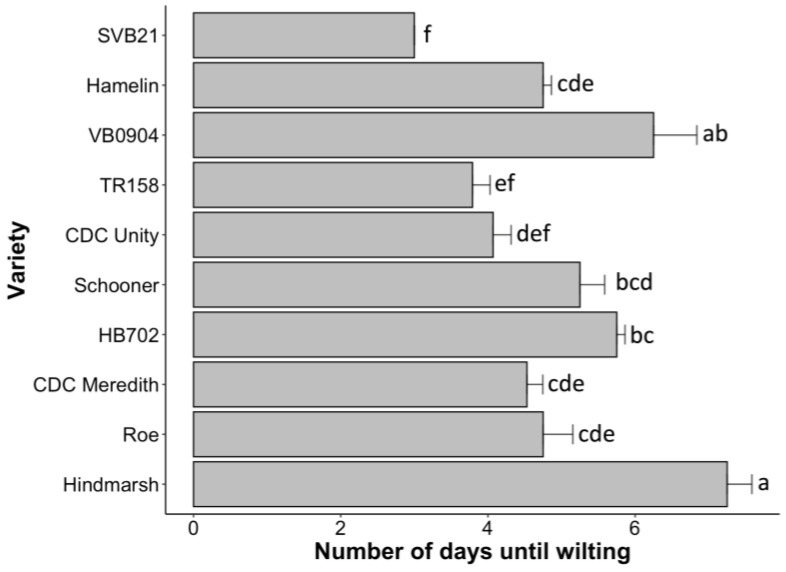
Bar plot illustrating the average number of days until signs of wilting were apparent for 10 varieties subjected to the withholding of water once the ZS49 stage was achieved. Varieties that significantly differ from one another are denoted by different letters, whereas those that do not significantly differ are denoted by the same letter. Statistical significance was attributed at *p* < 0.05.

**Figure 7 plants-12-02840-f007:**
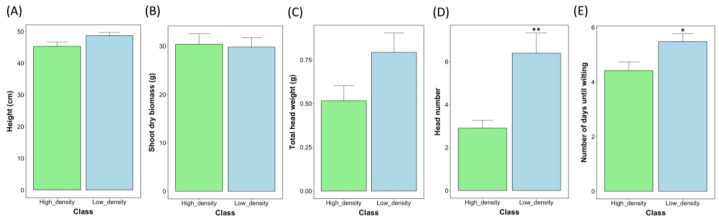
Panel figure highlighting the significant differences between low-stomatal-density and high-stomatal-density phenotypes for the traits of height (**A**), shoot dry biomass (**B**), head weight (**C**), head number (**D**), and the average number of days until wilting (**E**) during drought treatment. Statistical significance was attributed at *p* < 0.05. Significance levels are denoted by the following: * = *p* < 0.05; ** = *p* < 0.01. Standard error bars are also provided in the above barplots for each variety for the given traits.

**Figure 8 plants-12-02840-f008:**
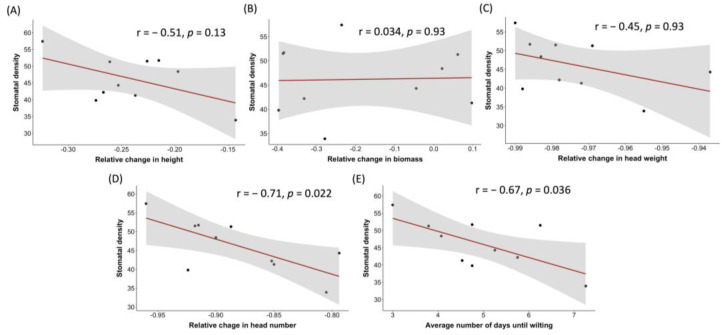
Panel figure comprising scatterplots depicting the relationship between the trait of stomatal density (*y*-axis) in relation to ‘relative change’ during the drought treatment (versus the control treatment) for 10 barley varieties for one of the following traits on the *x*-axis: height (**A**), shoot dry biomass (**B**), head weight (**C**), and head number (**D**). (**E**) depicts the relationship between stomatal density and the average number of days until wilting for the 10 barley varieties in this experiment. Also included on the plots are the linear regression line (red) and confidence bars (grey shaded areas). The Pearson correlation co-efficient (r) is provided, along with its associated *p*-value, in each of the respective plots. Significance was attributed at *p* < 0.05. Varieties were subjected to drought treatment at the ZS49 growth stage.

## Data Availability

Datasets utilised in this study are available upon reasonable request from the authors.

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
