# Peer review of "A Comparison of Different Stomatal Density Phenotypes of *Hordeum vulgare* under Varied Watering Regimes Reveals Superior Genotypes with Enhanced Drought Tolerance"

_plants, 2023, doi:10.3390/plants12152840_

Round 1
Reviewer 1 Report
In this article entitles “A comparison of different stomatal density phenotypes of Hordeum vulgare under varied watering regimes reveals superior genotypes with enhanced drought tolerance”, authors compared 10 barley varieties for the stomatal traits and try to link them with drought tolerance. The authors described detailed agronomy traits among the ten varieties in response to drought stress. They identified Hindmarsh as the best variety that exhibits superior biomass after drought treatment. However, there is lack of evidence to show mechanisms in this drought tolerance variety.
1. Please place Fig.1 after line 136 and shorthern caption description. Please revise the caption to “Figure 1. The average stomatal density of ten barley varieties. The fifth leaf at the tillering stage was recorded. Stomatal density was number of stomata/mm2 of leaf area. Error bars indicate the SD of the mean from 4(?) plants.
2. Use the lowercase letter of “r” for Pearson correlation coefficient not “R” in all text and Figures.
3. Line 228, change 637mm to 63.7cm, use cm for plant height in all text and Figures.
4. Line 462 and others, please change “improved biomass generation” to improved biomass “production”.
5. Line 823, “Pearson correlation coefficient (R)”, please use “r”.
Author Response
General comment: In this article entitles “A comparison of different stomatal density phenotypes of Hordeum vulgare under varied watering regimes reveals superior genotypes with enhanced drought tolerance”, authors compared 10 barley varieties for the stomatal traits and try to link them with drought tolerance. The authors described detailed agronomy traits among the ten varieties in response to drought stress. They identified Hindmarsh as the best variety that exhibits superior biomass after drought treatment. However, there is lack of evidence to show mechanisms in this drought tolerance variety.
General comment answer: We would like to thank the reviewer for their summary and helpful suggestions regarding the manuscript. Further clarification is needed in terms of reviewer highlighting a lack of evidence to show mechanisms for drought tolerance in Hindmarsh. Is this referring to the physiological basis of drought tolerance in Hindmarsh? If so, in our paper we highlight potential association of low stomatal density contributing to the Hindmarsh variety’s observed capacity for water retention under drought during the tillering stage (section 3.2), superior wilting resistance relative to all other varieties screened (section 3.4) and finally capacity for head generation under drought (section 3.4). Potential association of low stomatal density with drought tolerance was further evidenced by most severe drought impacts to yield traits being observed for variety SVB21 which exhibited the highest stomatal density in the study.
Comment 1: Please place Fig.1 after line 136 and shorten caption description. Please revise the caption to “Figure 1. The average stomatal density of ten barley varieties. The fifth leaf at the tillering stage was recorded. Stomatal density was number of stomata/mm2 of leaf area. Error bars indicate the SD of the mean from 4(?) plants.
Answer 1: We thank the reviewer for this suggestion and have moved Figure 1 accordingly, which improves the overall flow of the manuscript. The Figure 1 caption has also been amended as per the suggestion above to “The average stomatal density of ten barley varieties. The fifth leaf at the tillering stage was recorded. Stomatal density was number of stomata/mm2 of leaf area. Error bars indicate the standard error of the mean (n = 4 per variety).”
Comment 2: Use the lowercase letter of “r” for Pearson correlation coefficient not “R” in all text and Figures.
Answer 2: We have accordingly amended all instances of ‘R’ (referencing the Pearson correlation co-efficient) in text to reflect the lower case lettering ‘r’. Figure 8 has also been amended to reflect this change.
Comment 3: Line 228, change 637mm to 63.7cm, use cm for plant height in all text and Figures.
Answer 3: All instances of millimetre units for height measurements have been converted to cm in the main text. This change has also been reflected in panel ‘A’ of Figure 5 and panel ‘A’ of Figure 7.
Comment 4: Line 462 and others, please change “improved biomass generation” to improved biomass “production”.
Answer 4: In response, we have modified the manuscript and changed all instances of “improved biomass generation” to “improved biomass production.
Comment 5: Line 823, “Pearson correlation coefficient (R)”, please use “r”.
Answer 5: Modified line 823 to reflect the above suggested change form R to r.
Reviewer 2 Report
Revision to
Title: A comparison of different stomatal density phenotypes of Hordeum vulgare under varied watering regimes reveals superior genotypes with enhanced drought tolerance
Authors: Brittany Clare Robertson, Yong Han and Chengdao Li
Abstract Journal: Plants
Manuscript number: plants-2514528
General remarks: The manuscript by Robertson et al. investigates about the importance of stomatal density in ten different barley genotypes and the effects of these differences in term of drought tolerance. The manuscript is very interesting and reports suitable results identifying significant traits for the improvement of barley. In my opinion, the manuscript will be suitable for publication on Plants after minor revision.
Specific comments:
- In the introduction, authors describe very few information about the model organism Please detail the worldwide importance of barley and in specific regions and the general effects of drought on this crop.
- Authors should specify in the introduction (or in methods) the rationale about the selection of the ten varieties.
- Figure 1 did not show any significant difference. Is this correct?
- A general improvement of figure quality is necessary
- Is some part the manuscript is too long. Please resume.
No specific revisions are requested for the English language
Author Response
General comment: The manuscript by Robertson et al. investigates about the importance of stomatal density in ten different barley genotypes and the effects of these differences in term of drought tolerance. The manuscript is very interesting and reports suitable results identifying significant traits for the improvement of barley. In my opinion, the manuscript will be suitable for publication on Plants after minor revision.
General comment answer: We would like to thank the reviewer for providing suggestions on our manuscript. We have implemented the following changes, as per the reviewer’s suggestions below.
Comment 1: In the introduction, authors describe very few information about the model organism Please detail the worldwide importance of barley and in specific regions and the general effects of drought on this crop.
Answer 1: We agree that the barley crop was not sufficiently covered in the introduction section. In turn we have amended the introduction section to include information relating to the global economic importance of barley with some specific focus on Australia, which dominates the majority of the barley export market. In addition, key genetic mechanisms of barley that allow adaptation to abiotic stresses (including drought) is highlighted with some specific examples.
Comment 2: Figure 1 did not show any significant difference. Is this correct?
Answer 2: In response to the reviewer’s query, we confirm it is correct that Figure 1 does not show any significant differences between varieties for the trait of stomatal density on the fifth leaf. The main difference for stomatal density is in the mature leave (flag leave)
Comment 3: A general improvement of figure quality is necessary
Answer 3: Figures were modified as per reviewer’s request. Changes implemented to figures included increased font size of labels and axes and improved resolution of the new figures generated
Comment 4: Is some part the manuscript is too long. Please resume
Answer 4: In response to the reviewer’s comment elements of the manuscript were reduced in length where appropriate. The main body text of the manuscript is now under 10,000 words.
Additional comments post-revision from the authors
Following revision of the manuscript, the following errors were identified and have since been corrected in the revised version.
Section 2.1 in the results: The range of fifth leaf stomatal density measurements was incorrectly recorded as 5 stomata/mm2. This has now been corrected to 11 stomata/mm2 (line 141).
Section 2.4 in the results: The average wilting time for low stomatal density genotypes was incorrectly recorded as 5.39 days. This has been corrected to 5.48 days (line 609).
Figure 7, C: The average head weight for high stomatal density varieties (0.792g) was not plotted correctly. The bar plot has since been revised to reflect the average value of 0.792g.
Percentage difference values: There were some minor inconsistencies in the reporting of percentage difference across the traits studied under the various watering regimes resulting in discrepancies in the decimal values between some reported percentages. All percentage values have since been amended to be consistent throughout the manuscript.